# The DREAM Dataset: Supporting a data-driven study of autism spectrum disorder and robot enhanced therapy

Erik Billing[1]*, Tony Belpaeme[2,10], Haibin Cai[3], Hoang-Long Cao[4,8], Anamaria Ciocan[5], Cristina Costescu[5], Daniel David[5], Robert Homewood[1], Daniel Hernandez Garcia[2], Pablo Gómez Esteban[4,8], Honghai Liu[3], Vipul Nair[1], Silviu Matu[5], Alexandre Mazel[6], Mihaela Selescu[5], Emmanuel Senft[2], Serge Thill[1,9], Bram Vanderborght[4,8], David Vernon[1], Tom Ziemke[1,7]

1 University of Skövde, Skövde, Sweden, 2 University of Plymouth, Plymouth, United Kingdom, 3 University of Portsmouth, Portsmouth, United Kingdom, 4 Vrije Universiteit Brussel, Brussel, Belgium, 5 Universitatea Babeş-Bolyai, Cluj-Napoca, Romania, 6 SoftBank Robotics, Paris, France, 7 Linköping University, Linköping, Sweden, 8 Flanders Make, Lommel, Belgium, 9 Donders Institute for Brain, Cognition, and Behavior, Radboud University, Nijmegen, The Netherlands, 10 IDLab - imec, Ghent University, Ghent, Belgium

* erik.billing@his.se

**Data Availability Statement:** The complete dataset is available for download through the Swedish National Data Service, https://doi.org/10.5878/17p8-6k13. Sample data and usage instructions

## Abstract

We present a dataset of behavioral data recorded from 61 children diagnosed with Autism Spectrum Disorder (ASD). The data was collected during a large-scale evaluation of Robot Enhanced Therapy (RET). The dataset covers over 3000 therapy sessions and more than 300 hours of therapy. Half of the children interacted with the social robot NAO supervised by a therapist. The other half, constituting a control group, interacted directly with a therapist. Both groups followed the Applied Behavior Analysis (ABA) protocol. Each session was recorded with three RGB cameras and two RGBD (Kinect) cameras, providing detailed information of children's behavior during therapy. This public release of the dataset comprises body motion, head position and orientation, and eye gaze variables, all specified as 3D data in a joint frame of reference. In addition, metadata including participant age, gender, and autism diagnosis (ADOS) variables are included. We release this data with the hope of supporting further data-driven studies towards improved therapy methods as well as a better understanding of ASD in general.

## 1 Introduction

Children diagnosed with Autism Spectrum Disorder (ASD) typically suffer from widespread difficulties in social interactions and communication, and they exhibit restricted interests and repetitive behavior [1]. ASD is referred to as a spectrum disease because the type and the severity of the symptoms vary significantly between individuals. At one pole mild difficulties in social interaction and communication, such as problems in the initiation and maintenance of a conversation, the integration of verbal and nonverbal communication and the behavior adaptation to various contexts, together with some behavior rigidity can be seen. The opposite pole is characterized by

can be accessed at https://github.com/dream2020/data. In addition, source code for the DREAM RET System, with which the present dataset was gathered, is made available at https://www.doi.org/10.5281/zenodo.3571992.

**Funding:** This work was funded by the EU, under the Seventh Frame Programme grant #611391: Development of Robot-Enhanced therapy for children with AutisM spectrum disorders (DREAM). The commercial company SoftBank Robotics provided support in the form of salaries for author A.M., but did not have any additional role in the study design, data collection and analysis, decision to publish, or preparation of the manuscript. The specific roles of these authors are articulated in the 'author contributions' section.

**Competing interests:** The stuy from which the manuscript data was collected made use of the Nao robot, which is developed and sold by SoftBank Robotics with whom author A.M. is affiliated. This does not alter our adherence to PLOS ONE policies on sharing data and materials.

severe deficits in verbal and nonverbal communication, low level of social initiation, absence of peer interest, strong behavior inflexibility, and restricting/repetitive behaviors [1].

Behavioral and psychosocial interventions are the main approach for the treatment of ASD, while medication is sometimes prescribed in order to control associated symptoms or other comorbid problems [2]. Behavioral and psychosocial interventions try to facilitate the development and adaptation of children by teaching them appropriate social and communication skills, according to their developmental age. These interventions vary in terms of how structured the therapeutic activities are (e.g., during naturalistic play or following a pre-established activity), who delivers the intervention (e.g., a trained therapist or a parent), and the degree to which the child is required to follow a desired curriculum or the curriculum will be developed around child's preferences and interests [3, 4].

The therapeutic intervention that currently has the most consistent empirical support is Applied Behavior Analysis (ABA) [5]. ABA is a structured intervention following behavioral learning principles, in which reinforcements are manipulated in order to increase the frequency of desired behaviors and decrease the frequency for those that are maladaptive. The discrete trial training (DTT) is a common method employed by ABA treatments, in which the child is presented with a discriminative stimulus for a specific behavior (e.g., an instruction from the therapist), and the child receives a reward if he or she performs the expected behavior. If he or she does not, the therapist might correct the behavior by offering a demonstration or by offering a prompt [6]. In order to be effective, ABA therapies need to be both intensive and extensive and are thus associated with significant efforts from both patients and therapists providing the treatments.

An alternate form of therapy receiving a lot of attention over the last decade is Robot Assisted Therapy (RAT) [7–11], sometimes referred to as Robot Enhanced Therapy (RET) [12–16]. While RAT refers to a wide spectrum of approaches to autism therapy involving robots in one way or another, the notion of RET is used in a more narrow sense and refers to therapies following an ABA protocol where a humanoid robot constitutes an interaction partner. Both RAT and RET typically involve triadic interactions, comprising the child, a robot, and an adult, e.g., a therapist.

In RET interventions developed on ABA principles, the robot guides the child through a game-like activity in order to develop a behavior that is relevant for social communication, while the therapist supervises the interaction. The robot acts as a model by performing the desired behavior, or as a discriminant stimulus, by giving verbal or non-verbal instructions. The robot also acts as a source of social reinforcement, by providing positive or negative feedback on the performance of the child. The justification for using a robot in this form of treatment relies on the empirical findings indicating that ASD children are learning social behaviors from these interactions and might be more motivated to participate in the intervention as a result of the presence of the robot [17, 18].

Robots have also been proposed as a means for screening, diagnosis, and improved understanding of ASD [19, 20], the potential of which are still not fully exploited due to a majority of research on RAT and RET taking the form of small scale or single-case studies, without the methodological rigor required to make the data applicable in clinical domains [12, 21].

Within the European research project *DREAM—Development of Robot-Enhanced therapy for children with Autism spectrum disorders* [22], we have conducted a large scale clinical evaluation of RET, involving 61 children (9 female) between 3 and 6 years of age. 30 of the children interacted with a humanoid robot NAO [23] (RET-group), and the remaining 31 participants received a standard human treatment (SHT-group). The clinical efficacy of RET was tested in a randomized clinical trial design, with a study protocol consisting of an initial assessment, eight bi-weekly personalized behavioral interventions, and a final assessment. Each intervention targeted three social skills; imitation, joint-attention, and turn-taking.

All therapies were recorded using a sensorized intervention table able to record and interpret the child's behavior during the intervention (analyzing, for example, eye gaze, body movement, facial expression) [24]. The table was developed in order to inform the control of a supervised-autonomous robot used in RET [13], but was also used to support assessment and analysis of SHT. A total of 3121 therapy sessions was recorded, covering 306 hours of therapy.

While the clinical results from the evaluation are in the process of being published elsewhere, we here present a public release of the DREAM dataset, made available for download by the Swedish National Data Service, https://doi.org/10.5878/17p8-6k13. Following the ethical approval and agreements with caregivers, this public release does not comprise any primary data from the study. Primary data refers to direct measurements, e.g., video and audio recordings, of children in therapy. Instead, this public release comprises secondary data not revealing the identity of the children. Secondary data refers to processed measurements from primary data, including 3D skeleton reconstructions and eye-gaze vectors.

Further background on data-driven studies of autism and relevant datasets is presented in Section 2, followed by a presentation of the clinical evaluation from which this dataset was gathered (Section 3). Details of the DREAM dataset are provided in Section 4. Finally, the paper concludes with a discussion in Section 5.

## 2 Background

Diagnosis of ASD involves the assessment of the child behaviors considering their social initiations and responses, their joint attention episodes, their social play and their repetitive and stereotypic movements [1]. This involves, for example, attention to the patient's eye-gaze, face-expressions, and hand movements at specific points in time. While this is very difficult for a novice, therapists, knowing the protocol well, are trained to observe and identify these expressions.

Considering the large effort involved in its diagnosis and treatment of ASD, there is an urgent need to better understand the autism spectrum and to develop new methods and tools to support patients, caregivers, and therapists [25]. One initiative was made by Thabtah [26], who developed a mobile application for screening of ASD, based on DSM-5 [1] and two questionnaire based AQ and Q-CHAT screening methods [27, 28]. While this is far from the only mobile application for screening of ASD, we believe this initiative stands out by, in contrast to several other applications, being supported by published research and by sharing parts of the underlying databases publicly [29]. Such datasets, covering for example traits, characteristics, diagnoses and prognoses of individuals diagnosed with ASD could be important assets, and are still very rare.

Mobile applications could be excellent tools, for example during screening, not the least by being very accessible to the broader population. However, complete diagnosis and treatments require more information, and other forms of interaction, than what can be achieved with a mobile application. For example, coverage of the patient's behavior and social interactions are critical components for both improved understanding of ASD and development of new tools.

One example that clearly demonstrates the value of data-driven analysis of ASD is the work by Anzulewicz et al. [30]. The authors report a computational analysis of movement kinematics and gesture forces recorded from 82 children between 3 and 6 years old. 37 of these children were diagnosed with autism. The analysis revealed systematic differences in force dynamics within the ASD group, compared to the typically developed children included in the study. Unfortunately, this dataset has not been released publicly.

Moving outside the autism spectrum, there are a couple of relevant datasets focusing on social interaction. One such example is the Tower Game Dataset [31], comprising multimodal recordings from 39 adults engaged in a tower building game. A total of 112 annotated sessions were collected, with an average length of three minutes. It focuses specifically on rich

dyadic social interactions. Similar to the data-set presented here, the Tower Game Dataset contains body skeleton and eye-gaze estimates. Additionally, the dataset is manually annotated with so-called *Essential Social Interaction Predicates (ESIPs)*. The authors promise that *"a dataset visualization software [. . .] is available and will be released with the dataset"* [31], but unfortunately the dataset does not appear to be publicly available online.

Another dataset covering social interaction is the Multimodal Dyadic Behavior Dataset (MMDB) [32, 33]. This dataset comprises audio and video recordings from semi-structured play between one adult and one child in the age of 1 to 2 years. To date, 160 sessions of 5-minute interaction from 121 children have been released. Videos are annotated automatically for gaze shifts, smiling, play gestures, and engagement. An attractive aspect of this dataset is that the raw data streams are provided, including a rich set of 13 RGB cameras, one Kinect (RGBD) camera, 3 microphones, and 4 Affectiva Q-sensors for electrodermal activity and accelerometry, worn by both the adult and the child.

Focusing instead on human-robot interaction, the UE-HRI dataset [34] is a recent example. It includes audio and video recordings of 54 adult participants engaged in spontaneous dialogue with the social robot Pepper. The interactions took place in a public space, and include both one-to-one and multi-party interactions.

To our knowledge, the only public dataset covering children's interaction with robots is the PInSoRo dataset [35]. This dataset concerns typically developed children not associated with autism, but shares a similar ambition to support a data-driven study of interaction. PInSoRo covers 45 hours of RGB video recordings, 3D recordings of the faces, skeletal information, audio recordings, as well as game interactions. In addition, the dataset comprises manual annotations of, for example, task engagement, social engagement, and attitude.

In sum, several datasets related to the study of social interaction, human-robot interaction, and autism can be found in the literature. Some are also released publicly, but none of them reach the same size as the dataset we present here. While there are other datasets with a similar, or even richer, set of features, none of these cover children diagnosed with ASD. Under the label *Behavior Imaging*, Rehg et al. [36, p. 87] explicitly argue for the need for such a dataset:

> We believe this approach can lead to novel, technology-based methods for screening children for developmental conditions such as autism, techniques for monitoring changes in behavior and assessing the effectiveness of intervention, and the real-time measurement of health-related behaviors from on-body sensors that can enable just-in-time interventions.

Since children diagnosed with ASD are often sensitive to new clothing and wearable equipment, we consciously avoided on-body sensors. However, in other respects, we hope that the present work constitutes one important step towards a data-driven study of autism outlined by Rehg et al. [36].

## 3 Clinical evaluation and protocol

The clinical evaluation of RET, from which the present dataset is gathered, was conducted between March 2017 and August 2018 at three different locations in Romania. 76 children, age 3 to 6 years, were recruited to the study, out of which 70 met the inclusion criteria and were randomly assigned one of two conditions, RET or SHT. Participants in both groups went through a protocol of initial assessment, eight interventions, and a final assessment. The effect of the treatment was assessed using the Autism Diagnostic Observation Schedule (ADOS), in terms of the difference between the initial and final assessments [37]. Nine children did not continue the treatment beyond initial diagnosis, e.g., as a result of high skill performance,

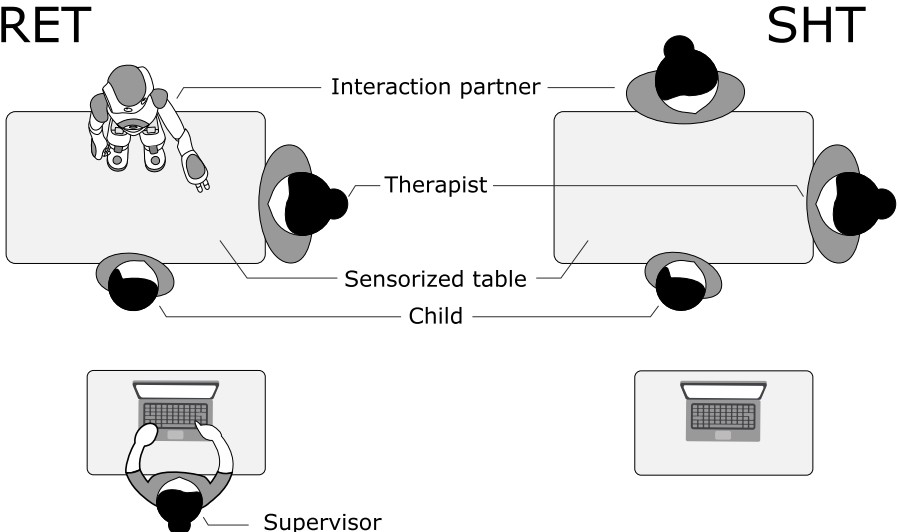

**Fig 1. Configuration of the therapy environment during the two conditions used.** The child interacts with either a humanoid robot (RET, left) or a therapist (SHT, right).

leaving 61 children with an initial ADOS score between 7 and 20 in the study (RET n = 30, SHT n = 31). A letter of consent was signed by at least one parent before initiating the study, expressing their consent to record the assessment and the intervention sessions and to use the data and recordings for scientific purposes in an anonymous fashion. The clinical study where this data was collected received prior ethical approval from the Scientific Council of Babes-Bolyai University in Cluj-Napoca, Romania, where the trial was conducted (record no. 30664/ February 10th, 2017). The clinical trial was pre-registered to the U.S. National Library of Medicine database (ClinicalTrials.gov) under the number NCT03323931.

The therapy environment followed two configurations illustrated in Fig 1. The two configurations (RET and SHT) were designed to be as similar as possible, with the interaction partner constituting the primary difference. A therapist was present during both conditions, seated at the side of the table. A picture from the RET condition is shown in Fig 2. The exact setup varied slightly between different tasks. Some tasks made use of a touch screen placed between the child and the interaction partner, referred to as a *sandtray* [38]. Other tasks had a table as illustrated in Fig 1.

Each intervention targeted three basic social skills that have been previously shown to be affected in individuals on the spectrum, namely *imitation* [39], *joint-attention* [40] and *turn-taking in collaborative play* [41]. The intervention had the same structure across all skills and was employed during both RET and SHT:

1. the interaction partner (robot or human) provided a discriminative stimulus (i.e., an instruction to perform a behavior that is relevant for a particular skill);

2. the interaction partner waited for the response of the child;

3. the interaction partner offered feedback that was contingent on the behavior of the child, namely a positive feedback if the behavior matched the one that was expected, or an indication to try again if the performance was below the expectation.

For each discriminative stimulus, the child had three attempts to perform the behavior, each trial following the same sequence from above. If the child failed to perform the behavior at the last attempt, then the therapist offered a behavioral prompt.

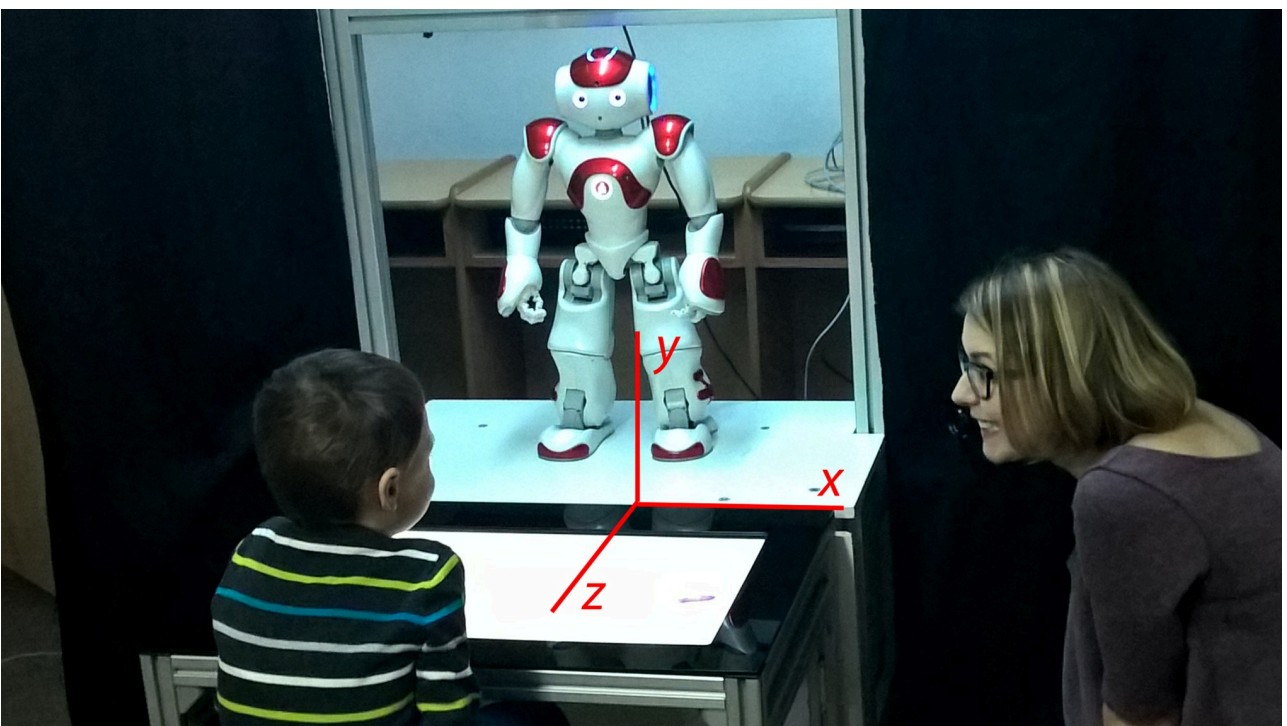

**Fig 2. Example of the therapy environment.** Red axes describe the orientation for the joint coordinate system for all data in the DREAM dataset.

Each intervention was divided into three to six parts, following a task script. This script specifies the task used during the intervention, instructions given to the child by the interaction partner (human or robot), as well as actions made by the interaction partner. In the RET condition, the robot follows the script automatically, supervised by a second therapist (supervisor) sitting behind the child (c.f. Fig 1). The supervisors role is to monitor the automatic interpretation of the child's behavior and to adjust the robot's responses if necessary. In the SHT condition, the supervisor is not present and the human interaction partner follows the script manually. Twelve unique intervention scripts were used, specifying different exercises and three difficulty levels. As the child reached maximum performance on one level, he/she moved to the next one.

For imitation there were three scripts: 1) imitation of objects (e.g., the child had to imitate a common way of playing with a toy car); 2) imitation of common gestures (e.g., waving hand and saying goodbye); and 3) imitation of gestures without a particular meaning (e.g., moving hand in a position that does not have any common reference). In each imitation script, the interaction partner performed the move first and asked the child to do the same. The child received positive feedback if he/she imitated accurately the behavior of the interaction partner.

For joint-attention there were also three levels of difficulty, varying by the number of cues offered by the interaction partner: 1) pointing and looking at an object placed in front of the child while also giving a verbal cue (i.e., "look"); 2) pointing and looking at an object without verbal cues; and 3) just looking at an object. The objects for this task were displayed as pictures on the sandtray placed in front of the child. The child received a positive feedback if he/she followed the cues and looked at the object indicated by the interaction partner.

For turn-taking there were three different types of tasks, each with two levels of difficulty. One task was focused on sharing information about what one likes most, by choosing from five pictograms that were displayed at once. The two levels differed by the complexity of the

pictograms (e.g., a simple color vs. an activity). Another task was focused on categorizing objects. In the first level of difficulty only one object that had to be categorized was displayed at a time, while in the second level there were eight such objects displayed, and the child had to choose one and move it in the correct category. The third task consisted of completing a series of pictures arranged in a pattern. In level one the child had to choose from two pictures the one that continues the pattern, while in the second level the child had to continue the pattern by choosing from four pictures. All turn-taking tasks were performed using the sandtray. The interaction partner and the child took turns in performing moves on the sandtray (e.g., choosing a favorite color). The child received a good performance rating and positive feedback if he/she waited without touching the screen while the interaction partner performed a move.

As mentioned above, the clinical protocol included an initial assessment, eight interventions, and a final assessment. The first and last assessments combined an ADOS evaluation with an evaluation of pre- and post-test performance in imitation, joint-attention and turn-taking. In the pre- and post-tests, the interaction partner did not provide any feedback, and the therapist did not offer any prompt (behavioral performance was only measured).

## 3.1 Sensors and setting

All therapy sessions were recorded using the same sensorized therapy table [24]. The table was equipped with three high-resolution RGB cameras and two RGBD (Kinect) cameras that, in combination with state of the art sensor interpretation methods, provide information about the child's position, motion, eye-gaze, face expressions, and verbal utterances. In addition, the table captured the presence and location of objects used in the therapy. A range of different algorithms were employed to compute these perceptions. A complete list of sensor primitives and associated methods is provided in Table 1. Note that only a subset of these features are included in the public dataset, see Section 4 for details.

The data gathered by the sensorized table was collected in real time with a temporal resolution of 25 Hz and was used to guide the behavior of the robot in the RET condition, following the session script. In addition, recorded data was used off-line to support analysis and assessment of the child's progress through therapy, in both RET and SHT conditions. The table works without sensors placed on the child and without individual calibration, both of which are problematic to employ in therapy with autistic children.

While sensor data was used to guide the robot's behavior on-line, robot responses were kept consistent throughout the study, i.e., the robot did not learn from previous interactions

**Table 1. Sensor primitives extracted by the sensorized intervention table.**

| Sensor primitive | Interpretation method |
|---|---|
| Relative eye-gaze | Two-eye model-based gaze estimation based on RGBD [42] |
| Head pose | Pose from Orthog-raphy and Scaling with ITerations (POSIT) [43] |
| Gaze estimation | A 3D gaze vector is achieved by combining the relative eye-gaze with calculated head pose [24] |
| Face detection | Boosted cascade face detector [44] |
| Facial features | Supervised descent method proposed by [45] |
| Face expressions | Frontalised Local Binary Patterns (LBP) classified using SVM [46] |
| 3D skeleton | Microsoft Kinect SDK |
| Action recognition | 3D joints Moving Trend method based on skeleton data [47] |
| Object tracking | GM-PHD Tracker [48] |
| Sound direction | Microsoft Kinect SDK |

Performance evaluations of each sensor primitive is available in Cai et al. [24].

with the child. Instead, suitable task difficulty was achieved through the session scripts as described above, combined with supervised autonomy ensuring reliable robot behavior even in cases when the system failed to correctly assess the child's actions [13]. While this architecture could effectively be combined with robot learning [49], here we chose a static system in order to increase validity of the clinical study.

## 4 Open dataset

The dataset resulting from the clinical evaluation presented above comprises a total of 306 hours of therapy. 41 out of 61 children finished all 8 interventions and the final diagnosis. The remaining participants finished an average of 4.5 interventions, i.e., just above half of the complete protocol. The total length of each intervention varied from a 3 to 87 minutes as a result of script length and child behavior, with a median duration of 32 minutes. Average intervention durations for each of the tow conditions (RET, SHT) are presented in Fig 3. To our knowledge, this is the largest dataset of autism therapy involving robots and probably also the largest recorded data set of children interacting with robots in general.

As mentioned in the introduction, this dataset does not comprise any direct measurements, i.e., raw data, from the conducted therapies. However, given that a RET framework involves processing of sensory data into higher level perceptions (see Section 3.1), we have access to comprehensive secondary data from these recordings. Any variables related to the clinical evaluation can however not yet be released, concerning for example the child's performances during therapy. This type of information may be included in a later release of the dataset. Other variables, such as face expressions, have relatively low reliability and were excluded from dataset for this reason. Finally, any variables that may reveal the child's identity have been excluded from the dataset.

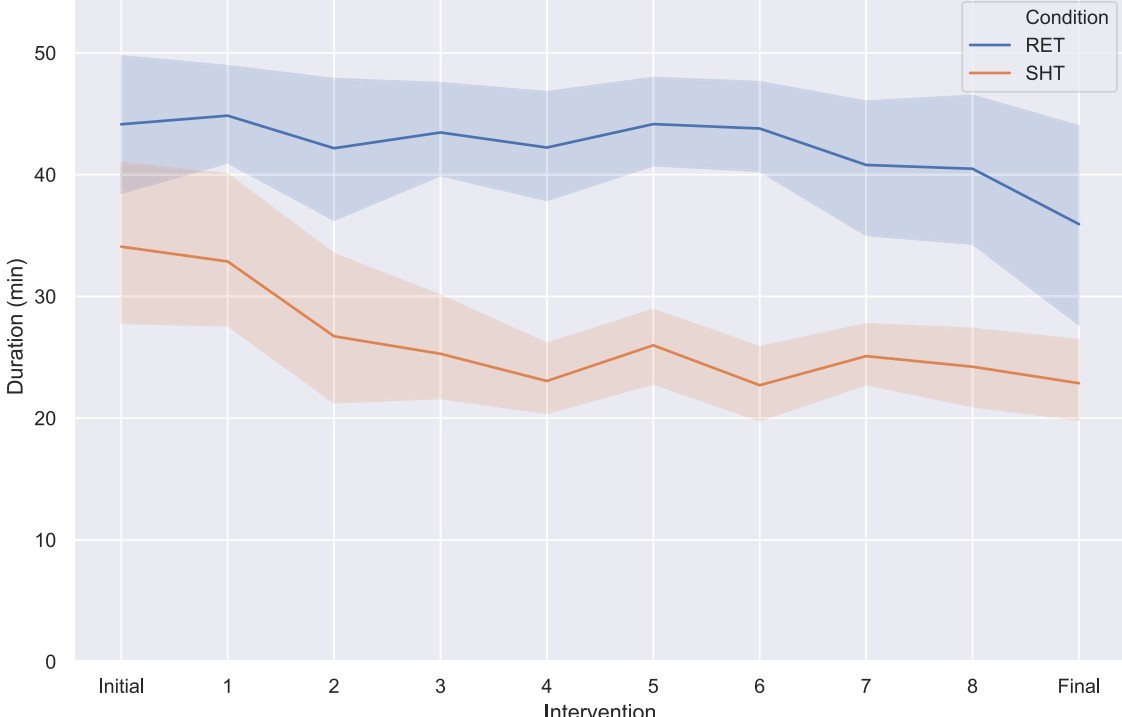

**Fig 3. Average duration of each intervention in the two conditions over the complete protocol, comprising initial assessment, 8 interventions, and a final assessment.** Envelopes represent the 95% confidence interval of the mean.

The following data has been selected for inclusion in the public release of the DREAM dxataset:

1. Child ID (numerical index),

2. Child's gender,

3. Child's age in months,

4. 3D skeleton comprising joint positions for upper body,

5. 3D head position and orientation,

6. 3D eye gaze vectors,

7. Therapy condition (RET or SHT),

8. Therapy task (Joint attention, Imitation, or Turn-taking),

9. Date and time of recording,

10. Initial ADOS scores.

With the ambition of releasing the dataset in an easily accessible, well-specified, and commonly used file format, JavaScript Object Notation (JSON) was selected. JSON is a stripped form of the JavaScript programming language, intended for data representation, https://www.json.org. JSON has many of the attractive attributes found in XML, including standard libraries for most programming languages, validation patterns, and human readability. However, JSON is less verbose than XML and includes standard notation for arrays, making it much more suitable for storing numeric data.

An example structure of the DREAM dataset JSON format is included in Table 2. The complete format is specified by the JSON Schema found in Appendix A. An important attribute of

**Table 2. Example structure of the open DREAM dataset, in JSON format.** *"[. . .]"* corresponds to numeric arrays that are too long to include here.

```
1  {
2    "$id": "User_37_18_Intervention_2_20171102_123242.369000.json",
3    "$schema": "https://raw.githubusercontent.com/dream2020/data/master/specification/dream.1.2.json",
4    "ados": {
5      "preTest": {
6        "communication": 2,
7        "interaction": 5,
8        "module": 1.0,
9        "play": 1,
10       "protocol": "ADOS-G",
11       "socialCommunicationQuestionnaire": 23,
12       "stereotype": 0,
13       "total": 7
14     }
15   },
16   "participant": {"id": 37, "gender": "male", "ageInMonths": 47},
17   "date": "2017-04-23T18:25:43.511Z",
18   "condition": "RET",
19   "task": {"index": 18, "ability": "TT","difficultyLevel": 1, start: 0, end: 10279},
20   frame_rate: 25.1,
21   "eye_gaze": {"rx": [...],"ry": [...],"rz": [...]},
22   "head_gaze": {"rx": [...],"ry": [...],"rz": [...]},
23   "skeleton": {
24     "elbow_left": {"x": [...],"y": [...],"z": [...],"confidence": [...]},
25     "elbow_right": {"x": [...],"y": [...],"z": [...],"confidence": [...]},
26     "hand_left": {"x": [...],"y": [...],"z": [...],"confidence": [...]},
27     "hand_right": {"x": [...],"y": [...],"z": [...],"confidence": [...]},
28     "head": {"x": [...],"y": [...],"z": [...],"confidence": [...]},
29     "sholder_center": {"x": [...],"y": [...],"z": [...],"confidence": [...]},
30     "sholder_left": {"x": [...],"y": [...],"z": [...],"confidence": [...]},
31     "sholder_right": {"x": [...],"y": [...],"z": [...],"confidence": [...]},
32     "wrist_left": {"x": [...],"y": [...],"z": [...],"confidence": [...]},
33     "wrist_right": {"x": [...],"y": [...],"z": [...],"confidence": [...]}
34   }
35 }
```

this dataset is that all attributes are defined in a common frame of reference using a Cartesian coordinate system. The orientation of the Cartesian space in relation to the therapy environment is visualized in Fig 2.

## 4.1 Licence

The Open DREAM dataset is licensed under

> **Creative Commons Attribution-NonCommercial-ShareAlike 4.0 International Licence** [50].

This licence permits *copying and redistribution of the material in any medium or format* and states that *the licensor cannot revoke these freedoms as long as you follow the licence terms*. The *material* here refers to all secondary data and metadata included in this public release of the dataset, excluding its underlying recordings or direct measurements. This freedom is given under the following terms:

1. **Attribution**—You must give appropriate credit, provide a link to the licence, and indicate if changes were made. You may do so in any reasonable manner, but not in any way that suggests the licensor endorses you or your use.

2. **NonCommercial**—You may not use the material for commercial purposes.

3. **ShareAlike**—If you remix, transform, or build upon the material, you must distribute your contributions under the same licence as the original.

4. **No additional restrictions**—You may not apply legal terms or technological measures that legally restrict others from doing anything the licence permits.

   **Note**:

- You do not have to comply with the licence for elements of the material in the public domain or where your use is permitted by an applicable exception or limitation.

- No warranties are given. The licence may not give you all of the permissions necessary for your intended use. For example, other rights such as publicity, privacy, or moral rights may limit how you use the material.

## 4.2 Data visualization

An important aspect of any public dataset is to make it accessible, and understandable, for a larger audience. In addition to the technical specification provided in Appendix A, we release a visualization tool named *DREAM Data Visualizer*. This tool runs directly in the web-browser and comprises a 3D environment where a user can playback each intervention, view the child's movements in relation to the interaction partner.

The DREAM Data Visualizer is released as open source under GNU GPL and available for download at github.com/dream2020/DREAM-data-visualizer. An example screenshoot from the DREAM data visualizer is presented in Fig 4.

## 5 Discussion

Although the global prevalence of autism is difficult to assess [51] and its exact origin has not been determined [5, 52], autism is becoming an increasingly common diagnosis worldwide. This affects not only all the people receiving diagnosis, but also constitutes a significant cost

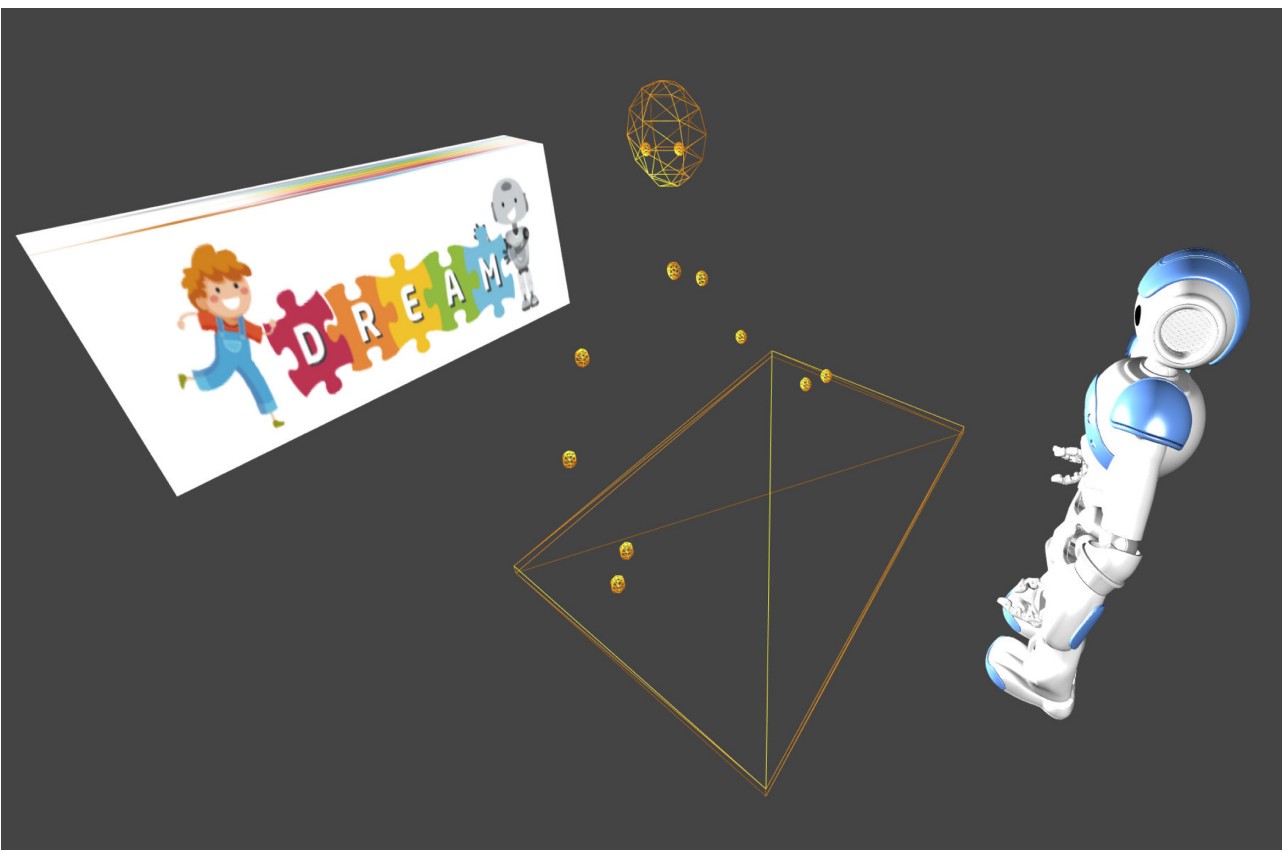

**Fig 4. Visualization of the DREAM dataset, including 3D skeleton of upper body and head rotations.**

for society [30]. RAT/RET has been put forward as a potentially cost-effective treatment, but still lacks large-scale clinical trials and longitudinal studies in order to assess its effects [12, 15, 17].

In the present work, we present a dataset covering behavioral data recorded during therapeutic interventions with 61 children. The dataset comprises a rich set of features which we believe are essential for understanding and assessing childrens' behavior during therapy. By providing a large set of data, comprising 61 children with varying degree of ASD taking part in more than 3000 therapy sessions, we hope that the DREAM Dataset can constitute an important asset in future studies in the field.

The dataset may for example be used by studies employing machine learning or artificial intelligence to find patterns in behavioral data. Such patterns may guide further clinical studies by providing new insights into how to appropriately select between RET and traditional ABA therapies or constitute input to new therapeutic methods.

## Data Availability

The complete dataset is available for download through the Swedish National Data Service, https://doi.org/10.5878/17p8-6k13. Sample data and usage instructions can be accessed at https://github.com/dream2020/data. In addition, source code for the DREAM RET System [24], with which the present dataset was gathered, is made available at https://www.doi.org/10.5281/zenodo.3571992.

## Appendix A: Dataset specification

This is a JSON schema for the open DREAM dataset presented in Section 4. The schema presented here specifies the format for a single therapy session with one child under therapy, including definitions of all mandatory attributes of the data. A JSON database file may however comprise additional attributes not defined here. The complete dataset comprises a large set of these sessions.

```
1   {
2     "$schema": "http://json-schema.org/draft-07/schema#",
3     "$id": "http://dream2020.eu/data/specification/dream.1.2.json",
4     "title": "DREAM dataset format specification",
5     "description": "This is a JSON schema for the DREAM dataset, an outcome of a European research project DREAM: Development of Robot-
                       Enhanced therapy for children with AutisM spectrum disorders. More info at http://dream2020.eu.",
6     "type": "object",
7     "properties": {
8       "date": {"type":"string","format":"date-time"},
9       "ados": {
10        "type": "object",
11        "properties": {"initial": {"type": "number"}}
12      },
13      "condition": {"type": "string","pattern": "^(RET|SHT)$"},
14      "task": {
15          "type": "object",
16          "properties": {
17              "index": {"type": "integer"},
18              "ability": {"type": "string"},
19              "difficultyLevel": {"type":"integer"},
20              "start": {"type": "integer"}, "end": {"type": "integer"}
21          }
22      },
23      "participant": {
24          "type": "object",
25          "properties": {
26            "id": {"type": "integer"},
27            "gender": {"type": "string","pattern": "^(male|female)$"},
28            "age": {"type": "number","minimum": 0}
29          },
30          "required": ["id"]
31      },
32      "frame_rate": {"type": "number"},
33      "eye_gaze": {"$ref": "#/definitions/rot"},
34      "head_gaze": {"$ref": "#/definitions/rot"},
35      "skeleton": {
36          "type": "object",
37          "properties": {
38            "elbow_left": {"$ref": "#/definitions/pos"},
39            "elbow_right": {"$ref": "#/definitions/pos"},
40            "hand_left": {"$ref": "#/definitions/pos"},
41            "hand_right": {"$ref": "#/definitions/pos"},
42            "head": {"$ref": "#/definitions/pos"},
43            "sholder_center": {"$ref": "#/definitions/pos"},
44            "sholder_left": {"$ref": "#/definitions/pos"},
45            "sholder_right": {"$ref": "#/definitions/pos"},
46            "wrist_left": {"$ref": "#/definitions/pos"},
47            "wrist_right": {"$ref": "#/definitions/pos"}
48          },
49          "required": [
50            "elbow_left","elbow_right","hand_left","hand_right","head","sholder_center","sholder_left","sholder_right","wrist_left","
                 wrist_right"
51          ]
52        }
53    },
54    "required": ["eye_gaze","head_gaze","skeleton","participant","ados","condition","task","frame_rate"],
55    "definitions": {
56      "pos": {
57        "$id": "#pos",
58        "description": "Absolute position in Cartesian space.",
59        "type": "object",
60        "properties": {
61          "x": {"type": "array","items": {"type": ["number","null"]}},
62          "y": {"type": "array","items": {"type": ["number","null"]}},
63          "z": {"type": "array","items": {"type": ["number","null"]}},
64          "confidence": {"type": "array","items": {"type": ["number","null"]}}
65        },
66        "required": ["x","y","z","confidence"]
67      },
68      "rot": {
69        "$id": "#rot",
70        "description": "Cartesian heading vector.",
71        "type": "object",
72        "properties": {
73          "rx": {"type": "array","items": {"type": ["number","null"],"minimum": -1,"maximum": 1}},
74          "ry": {"type": "array","items": {"type": ["number","null"],"minimum": -1,"maximum": 1}},
75          "rz": {"type": "array","items": {"type": ["number","null"],"minimum": -1,"maximum": 1}}
76        },
77        "required": ["rx","ry","rz"]
78      }
79    }
80  }
```

## Author Contributions

**Conceptualization:** Erik Billing, Honghai Liu, Alexandre Mazel, Serge Thill, Bram Vanderborght, Tom Ziemke.

**Data curation:** Erik Billing, Robert Homewood, Honghai Liu, Vipul Nair, Emmanuel Senft.

**Funding acquisition:** Daniel David, Honghai Liu, Serge Thill, Bram Vanderborght, David Vernon, Tom Ziemke.

**Investigation:** Haibin Cai, Anamaria Ciocan, Cristina Costescu, Daniel David, Silviu Matu, Mihaela Selescu.

**Methodology:** Tony Belpaeme, Haibin Cai, Hoang-Long Cao, Cristina Costescu, Daniel David, Silviu Matu, Serge Thill, Bram Vanderborght, David Vernon.

**Project administration:** Erik Billing, Tony Belpaeme, David Vernon, Tom Ziemke.

**Resources:** Alexandre Mazel.

**Software:** Erik Billing, Haibin Cai, Hoang-Long Cao, Daniel Hernandez Garcia, Pablo Gómez Esteban, Honghai Liu, Emmanuel Senft, David Vernon.

**Supervision:** Erik Billing, Tony Belpaeme, Daniel David, Honghai Liu, Tom Ziemke.

**Validation:** Erik Billing, Anamaria Ciocan, Daniel David, Vipul Nair, Silviu Matu, Emmanuel Senft.

**Visualization:** Erik Billing, Robert Homewood.

**Writing – original draft:** Erik Billing.

**Writing – review & editing:** Tony Belpaeme, Daniel Hernandez Garcia, Honghai Liu, Emmanuel Senft, Serge Thill, Bram Vanderborght, Tom Ziemke.

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
