## [Decision Letter · Decision Letter 0]

19 May 2020

PONE-D-20-08566

The DREAM Dataset: Supporting a data-driven study of autism spectrum disorder and robot enhanced therapy

PLOS ONE

Dear Dr. Billing,

Thank you for submitting your manuscript to PLOS ONE. After careful consideration, we feel that it has merit but does not fully meet PLOS ONE’s publication criteria as it currently stands. Therefore, we invite you to submit a revised version of the manuscript that addresses the points raised during the review process.

We would appreciate receiving your revised manuscript by Jun 29 2020 11:59PM. To enhance the reproducibility of your results, we recommend that if applicable you deposit your laboratory protocols in protocols.io, where a protocol can be assigned its own identifier (DOI) such that it can be cited independently in the future. For instructions see: http://journals.plos.org/plosone/s/submission-guidelines#loc-laboratory-protocols

We look forward to receiving your revised manuscript.

Kind regards,

Lucia Billeci

Academic Editor

PLOS ONE

Journal Requirements:

'EU FP7 Grant #611391: Development of Robot-Enhanced therapy for children with AutisM spectrum disorders (DREAM).'

We note that one or more of the authors are employed by a commercial company: SoftBank Robotics

Reviewers' comments:

Reviewer's Responses to Questions

**Comments to the Author**

1. Is the manuscript technically sound, and do the data support the conclusions?

Reviewer #1: Yes

Reviewer #2: Yes

Reviewer #3: Yes

Reviewer #4: Yes

2. Has the statistical analysis been performed appropriately and rigorously? 

Reviewer #1: N/A

Reviewer #2: N/A

Reviewer #3: N/A

Reviewer #4: N/A

3. Have the authors made all data underlying the findings in their manuscript fully available?

Reviewer #1: Yes

Reviewer #2: Yes

Reviewer #3: Yes

Reviewer #4: Yes

4. Is the manuscript presented in an intelligible fashion and written in standard English?

Reviewer #1: Yes

Reviewer #2: Yes

Reviewer #3: Yes

Reviewer #4: Yes

5. Review Comments to the Author

Reviewer #1: Thank you for the opportunity to review this very interesting manuscript entitled “The DREAM Dataset: Supporting a data-driven study of autism spectrum disorder and robot enhanced therapy” which has been submitted for consideration in the Plos One. Robot Enhanced Therapy attracted great attention. I agree the importance of Robot Enhanced Therapy and dataset supporting further data-driven studies toward improved therapy methods as well as better understanding of ASD in general. However, I could not find any result in this article. In addition Discussion is very short.

Reviewer #2: The MS describes a database/set corresponding to a study involving a robot (robot enhanced therapy) in children with ASD. Authors have recorded many features during 3000 sessions and they offer a data visualizer that is very welcome. Given the unique dataset they are describing, authors should be congratulated for such a commitment.

However they are many imprecisions and writing issues that make the manuscript inadequate for publication. Authors need to work on it a bit more (It looks like a conference paper that we often have in the field).

1. The intro does not stand as it is. It should be reorganized and some points need to be added or modified.

I suggest:

- start by ASD treatment principles (see Narzisi et al. 2015).

- Don’t start by medication!! It is not the treatment of autism.

- The claim that medication has strong evidence is wrong! The statements on medication are useless.

- Then be more specific with ABA (because) it inspired your robot enhanced therapy.

- Add a brief paragraph on robot and ASD (they are several recent reviews).

- Then present RET and indicate in the intro the exact design (randomized?, duration?)

- Page 2 correct form for from

2. The clinical evaluation

- This section should be renamed “clinical evaluation and protocol”

- The most important revision should be done when authors describe the setting in this section. To me, it should be a separate section called “sensors and setting”.

- In this new section, the authors should detail more how the robot is used (at least grossly as I understand that details would be available in the clinical paper). Also they should offer a table with the extracted features and made available in the dataset (and the corresponding algorithm they used). Also they should indicate whether some raw data are availables.

3. The section open data set needs a 4.1 “dataset variables”. Also the 4.2 “licence” seems to be a very general statement found on websites! Please if you give a note be more specific in the case of Dream data set. No need of general legal statements!

4. The discussion is minimalist and why not, I am OK. But please edit the section as there are at least 2 sentences that I did not understood.

Also, please cite more recent reviews instead of [11, 14]. Finally, the statement very general that the data set will offer opportunity to develop new screening method makes no sense to me. You don’t have typical developing controls!

Reviewer #3: This is really impressive work and I commend the authors on what they have achieved in this project. Such a through dataset is a valuable contribution to the field and will undoubtedly be useful to other researchers. My only suggestion is that the authors provide more details on the duration of the therapy sessions in their overview. Currently the authors say that sessions lasted anywhere between a few minutes and 40 minutes. It would be useful to have a graph batching the sessions and providing an indication the average session might last and also if theses session durations increased or decreased over the course of the study.

Overall great work!

Reviewer #4: In this paper, the authors present a dataset of behavioral data recorded through a Robot Enhanced Therapy (RET) with 61 children diagnosed with Autism Spectrum Disorders (ASD). Half of the children interacted with the social robot NAO supervised by a therapist. The other half, constituting a control group, interacted directly with a therapist. Both groups followed the Applied Behavior Analysis (ABA) protocol. The dataset comprises body motion, head position and orientation, and eye gaze variables recorded with three RGB cameras and two RGBD (Kinect) cameras. It also includes metadata such as participant age, gender, and autism diagnosis (ADOS) variables. Participants in both groups went through a protocol of initial diagnosis, eight interventions, and a final diagnosis, targeting three social skills; Turn-Taking (TT), Imitation (IM) and Joint Attention (JA). The effect of the treatment was assessed using ADOS, in terms of the difference between the initial and final diagnosis. The clinical study where this data was collected received prior ethical approval from the Scientific Council of Babes-Bolyai University in Cluj-Napoca, Romania, where the trial was conducted (record no. 30664/February 10th, 2017). A letter of consent was signed by at least one parent before initiating the study.

I am very happy that this type of datasets is being publicly released. This is a very important step to help future robot-assisted therapies in the case of autism. I would like to congratulate the authors for this work.

I have only a few comments/questions and minor corrections suggested below.

Minor points

About the content of the dataset itself: Page 8, in the list of included data, I would replace ‘date’ by ‘time’, so that it is clear that not only the day is specified. Shouldn’t the success of the current trial also be indicated? This could help users of the dataset distinguish correct from incorrect movements, and maybe better understand cases of hesitation for instance. Moreover, the authors put an emphasis in the introduction on the different levels of difficulties/deficits of people within the ASD spectrum. Shouldn’t the dataset provide information distinguishing these different levels?

Lines 19:23, the sentence is ill-formulated: ‘[5] labeled two interventions as well-established: individual, comprehensive Applied Behavior Analysis (ABA) and teacher-implemented, focused, ABA+ developmental social-pragmatic (DSP), an intervention that combines ABA and DSP strategies.’ When I read the sentence, it seems to me that there are three interventions: ABA, DSP and ABA+. While actually there are ABA and ABA+DSP. And the way the sentence is written could suggest that DSP is an intervention that combines ABA and DSP. I suggest reformulating and separating the interventions with semicolons. Also, rather than following the way they are presented in the abstract of Smith & Iadarola 2015, it seems to me that it would be better to first list ABA and DSP, with a short explanation of what is the difference. And then explain that a combination of the two called ABA+DSP also exists. Then the authors could explain that Smith & Iadarola 2015 have emphasized their difference in efficacy. I am not sure it really useful to mention the distinction between focused and comprehensive, individual vs. teacher-implemented, since this is not further explained nor used in the present manuscript.

Lines 33-34: ‘an ABA protocol where a humanoid robot constitutes the interaction partner.’ I think ‘the’ is not appropriate here. Because it suggests at first glance that the robot is the unique interaction partner. I would replace with ‘an’. At the end of the paragraph, I think it would great to emphasis that in RET the goal is not to replace the human therapist by a robot, but instead to assist the therapist, the robot being only a mediator (of the therapy, as opposed to the therapist being a mediator of the interaction with the robot) or a tool.

I think it is important to state that the robot’s behavior is preprogrammed, not allowing any on-the-fly learning while interacting with the child. The strengths of doing so could be emphasized, such as stability, predictability, perhaps easier acceptability by children with ASD, and making sure that the behavior of the robot during the experiment is perfectly controlled. In contrast, it would interesting to mention that alternative studies enable the robot to learn on-the-fly while interacting with children with autism. The strengths and weaknesses of doing so could also be discussed in comparison with the present method. I think this would be very useful, first so that potential users of the dataset know clearly what was the robot’s behavior and its abilities, and second to provide some insights to the community about the pros and cons of enabling robots to learn or not during RAT/RET.

Page 3/16, the authors should not forget to remove the last three sentences of footnote 2 before publication.

One the one hand, the background section stresses the potential of this kind of dataset to contribute to the diagnosis of ASD. On the other hand, the introduction mentions only therapies, but not the use of robots in diagnosis. I think the objectives should be more clearly stated, and the potential contribution to therapy, diagnosis or both should be discussed. This of particular interest for the social robotics community which is currently wondering whether there is a potential for social robots to contribute to therapy only, or also to diagnosis.

Line 98, ‘It focus specifically’ -> It focuses /or/ Its focus is.

Figure 1 seems to suggest that no supervising human was involved in the SHT configuration. Could the authors confirm? Did the supervisor have an active role (like intervention), in addition to controlling the robot in case of problem, or only a passive role (monitoring)? In the former case, was it a problem not to have a supervisor during SHT? Was there a difference between RET and SHT in terms of interventions by the supervisor?

Figure 2 does not explicitly refer to any touchscreen between the child and the robot. The sandtray is actually not clearly visible in Figure 2, in contradiction with what is written (lines 164-166).

Line 170: ‘were employed’ -> was employed.

Line 206, objected -> object.

Lines 231-232: ‘state of the art sensor interpretation algorithms’. Could the authors more explicitly state which algorithms were used? I think it is important for people using the dataset to know.

What happened when children only spent ‘a few minutes’ in a session? Did he/she complete only a single intervention script? Was the script completed? Was the data still included in the dataset?

Line 313, ‘screen shoot’ -> screenshot.

Line 327, ‘be used studies’ -> be used by studies.

Line 352, comprise -> comprises.

6. PLOS authors have the option to publish the peer review history of their article (what does this mean?). If published, this will include your full peer review and any attached files.

Reviewer #1: No

Reviewer #2: No

Reviewer #3: No

Reviewer #4: Yes: Mehdi Khamassi

---

## [Author Response · Author response to Decision Letter 0]

25 Jun 2020

Dear Prof Kovacs, 

thanks for looking over the revised manuscript. I've now removed all figures in the manuscript but left the figure labels/texts in order to keep in-text figure references consistent. I hope this update is satisfactory. 

Please find updated statements regarding financial support and competing of interests in the cover letter, and a detailed response to all reviewer comments in the attached rebuttal letter. We've attached the revised manuscript in two versions, with and without change tracking.

Thanks again for a valuable input to the article and I wish you a good summer! 

Kind regards, Dr. Erik Billing on behalf of all the authors.

---

## [Decision Letter · Decision Letter 1]

17 Jul 2020

The DREAM Dataset: Supporting a data-driven study of autism spectrum disorder and robot enhanced therapy

PONE-D-20-08566R1

Dear Dr. Billing,

We’re pleased to inform you that your manuscript has been judged scientifically suitable for publication and will be formally accepted for publication once it meets all outstanding technical requirements.

Kind regards,

Lucia Billeci

Academic Editor

PLOS ONE

Additional Editor Comments (optional):

Reviewers' comments:

Reviewer's Responses to Questions

**Comments to the Author**

1. If the authors have adequately addressed your comments raised in a previous round of review and you feel that this manuscript is now acceptable for publication, you may indicate that here to bypass the “Comments to the Author” section, enter your conflict of interest statement in the “Confidential to Editor” section, and submit your "Accept" recommendation.

Reviewer #2: (No Response)

Reviewer #3: All comments have been addressed

Reviewer #4: All comments have been addressed

2. Is the manuscript technically sound, and do the data support the conclusions?

Reviewer #2: (No Response)

Reviewer #3: Yes

Reviewer #4: Yes

3. Has the statistical analysis been performed appropriately and rigorously? 

Reviewer #2: (No Response)

Reviewer #3: Yes

Reviewer #4: Yes

4. Have the authors made all data underlying the findings in their manuscript fully available?

Reviewer #2: (No Response)

Reviewer #3: Yes

Reviewer #4: Yes

5. Is the manuscript presented in an intelligible fashion and written in standard English?

Reviewer #2: (No Response)

Reviewer #3: Yes

Reviewer #4: Yes

6. Review Comments to the Author

Reviewer #2: (No Response)

Reviewer #3: After reviewing the paper I am happy that the authors have made the necessary changes to the manuscript and it is acceptable to publish in its current form.

Reviewer #4: (No Response)

7. PLOS authors have the option to publish the peer review history of their article (what does this mean?). If published, this will include your full peer review and any attached files.

Reviewer #2: No

Reviewer #3: No

Reviewer #4: **Yes: **Mehdi Khamassi

---

## [Editor Report · Acceptance letter]

28 Jul 2020

PONE-D-20-08566R1 

The DREAM Dataset: Supporting a data-driven study of autism spectrum disorder and robot enhanced therapy 

Dear Dr. Billing:

I'm pleased to inform you that your manuscript has been deemed suitable for publication in PLOS ONE. Congratulations! Your manuscript is now with our production department. 

Kind regards, 

on behalf of

Dr. Lucia Billeci 

Academic Editor

PLOS ONE